# Sequential Cascade Doping of Conjugated-Polymer-Wrapped Carbon Nanotubes for Highly Electrically Conductive Platforms

**DOI:** 10.3390/polym16131884

**Published:** 2024-07-01

**Authors:** Da Young Lee, Da Eun Choi, Yejin Ahn, Hyojin Kye, Min Seon Kim, Bong-Gi Kim

**Affiliations:** 1Department of Materials Science and Engineering, Konkuk University, 120 Neungdong-ro, Seoul 05029, Republic of Korea; flowerfull@konkuk.ac.kr; 2Department of Organic and Nano System Engineering, Konkuk University, 120 Neungdong-ro, Seoul 05029, Republic of Korea; daeun5467@gmail.com (D.E.C.); alslzl1129@naver.com (Y.A.); hjk01109@konkuk.ac.kr (H.K.); mskim2023@kitech.re.kr (M.S.K.)

**Keywords:** conjugated polymer, polymer-SWCNT hybridization, cascade doping, doping mechanism, flexible conductor

## Abstract

To explore a highly conductive flexible platform, this study develops PIDF-BT@SWCNT by wrapping single-walled carbon nanotubes (SWCNTs) with a conjugated polymer, PIDF-BT, known for its effective doping properties. By evaluating the doping behaviors of various dopants on PIDF-BT, appropriate dopant combinations for cascade doping are selected to improve the doping efficiency of PIDF-BT@SWCNT. Specifically, using F4TCNQ or F6TCNNQ as the first dopant, followed by AuCl_3_ as the second dopant, demonstrates remarkable doping efficiency, surpassing that of the individual dopants and yielding an exceptional electrical conductivity exceeding 6000 S/cm. Characterization using X-ray photoelectron spectroscopy and Raman spectroscopy elucidates the doping mechanism, revealing an increase in the proportion of electron-donating atoms and the ratio of quinoid structures upon F4TCNQ/AuCl_3_ cascade doping. These findings offer insights into optimizing dopant combinations for cascade doping, showcasing its advantages in enhancing doping efficiency and resulting electrical conductivity compared with single dopant processes.

## 1. Introduction

Flexible conductive platforms have recently drawn significant attention due to their potential applications in free-form optoelectronic devices, such as solar cells [1,2,3], field-effect transistors (FET) [4,5,6], thermoelectric devices [7,8,9], and wearable electronics [10,11,12]. Conjugated polymers (CPs) have been extensively investigated as a representative example of flexible conductive platforms because their electrical conductivity can be easily enhanced through chemical doping [13,14,15,16]. Solution-processible CPs have demonstrated high electrical conductivity exceeding 600 S/cm through efficient charge carrier generation by molecular-level dopants without using strong acids [17,18]. However, despite noticeable improvements, the conductivity of CP-based platforms remains unsatisfactory, significantly lower than that of conventional conductive ceramics and metals. This indicates that although ceramics and metals are less flexible compared to CPs, the electrical conductivity of CP-based conductors needs further enhancement for the stable operation of flexible electronic devices.

Carbon nanotubes (CNTs) have excellent electrical properties, such as high intrinsic carrier mobility reaching up to 10^5^ cm^2^/V·s [19]. Therefore, despite their limited solution processability due to strong cohesive interactions, various studies are being conducted to harness the high electrical properties and large surface area of CNTs [20,21,22]. In a previous study, we introduced CP-wrapped CNTs that exhibit excellent electrical conductivity of over 5000 S/cm with good dispersion characteristics [23]. The high conductivity of the CP-CNT core–shell material is attributed to the synergistic effects of the high mobility of CNTs and the carrier generation of outer CP chains enhanced by additional doping with small molecular dopants. Doping reduces the charge transfer barrier between CP and CNT, facilitating the use of high-mobility CNTs as the principal carrier transport channel. Indeed, a 4-probe FET fabricated with CP-wrapped single-walled CNTs (SWCNTs) revealed that charge carrier mobility is sensitively affected by the doping level. The mobility reached up to 138.0 cm^2^/V·s after AuCl_3_ doping, significantly higher than that of common CPs. Therefore, further increases in electrical conductivity can be expected through the application of a doping process that maximizes the doping efficiency of the polymer, as additional charge carriers created in the CPs by the doping process directly contribute to increasing the conductivity of CP-wrapped CNTs.

Various studies have been conducted on doping processes that can improve the doping efficiency of CPs. For example, Kiefer et al. achieved higher doping efficiency by sequentially applying two different dopants with shallow and relatively deep lowest unoccupied molecular orbital (LUMO) levels compared to using each single dopant individually [24]. Additionally, Yamashita et al. found that the doping efficiency of CPs can be further improved after initial doping with a kinetically favorable dopant, followed by efficient anion exchange with a second dopant [15]. Recently, we demonstrated that during cascade doping, the initially doped CPs release their primary dopant through anion exchange with a dopant possessing a deeper LUMO level [25]. This anion exchange process occurs via a transition state that requires relatively lower activation energy compared to doping with individual dopants. Consequently, higher doping efficiency can be achieved through anion exchange due to the additional participation of the released dopant and consecutive anion exchange. To maximize the efficiency of the sequential cascade doping, the first dopant should exhibit kinetically favorable doping characteristics, while the second dopant should possess an energetically favorable deeper LUMO level than the first dopant to facilitate spontaneous anion exchange.

In this study, using an indoloindole-based CP (PIDF-BT) and various dopants with verified doping characteristics [26,27,28,29,30,31], we specified the doping kinetics of each dopant by comparing the doping time required for the electrical conductivity to reach saturation. Based on preliminary screening of dopants for PIDF-BT, 2,3,5,6-tetrafluoro-7,7,8,8-tetracyanoquinodimethane (F4TCNQ) and AuCl_3_ were selected as principal dopants for their kinetically and energetically favorable doping properties, respectively. PIDF-BT@SWCNT was then prepared by wrapping PIDF-BT around commercially available SWCNTs, and the doping efficiency and resulting electrical conductivity were compared between individual dopant treatments and cascade doping. Additionally, the increased doping efficiency of the cascade doping process was verified through changes in UV–Vis absorbance, X-ray photoelectron spectroscopy (XPS), and electron spin resonance (ESR) measurements.

## 2. Results and Discussion

PIDF-BT and its analogues containing indoloindole moieties have demonstrated excellent doping abilities through efficient intermolecular charge transfer to dopants [8,13,32]. Additionally, F4TCNQ and 2,2′-(perfluoronaphthalene-2,6-diylidene) dimalononitrile (F6TCNNQ) exhibit excellent doping properties for CPs [26,27]. As the doping mechanism of F4TCNQ and F6TCNNQ is based on electron donor–acceptor interactions with electron-donating CPs, they are considered representative examples of molecular dopants [32]. Moreover, Lewis acids have been intensively investigated as alternatives to molecular dopants because they can create hole-type charge carriers in CPs through Lewis acid–base interactions [33,34]. Representative examples of efficient Lewis acids include nitrosonium tetrafluoroborate (NOBF_4_), AuCl_3_, FeCl_3_, and tris(4-bromophenyl)ammoniumyl hexachloroantimonate (Magic Blue), which have been adopted for CP doping [26,27,28,29,30,31]. Therefore, the doping behavior of PIDF-BT with each individual dopant was evaluated to identify suitable dopant combinations for sequential cascade doping in this study. The chemical structures of the materials used to confirm the doping characteristics of PIDF-BT are summarized in Figure 1a. The doping kinetics for each dopant were estimated through changes in electrical conductivity and UV–Vis absorbance over time. As displayed in Figure 1b, when comparing the time required for the saturation of electrical conductivity by each dopant in PIDF-BT doping, molecular dopants (F4TCNQ and F6TCNNQ) completed doping within 10 s, while Lewis acid dopants took longer than 100 s. This result indicates that molecular dopants are advantageous for CP doping compared to Lewis acid dopants, in terms of doping kinetics.

The doping characteristics of PIDF-BT over time can be clearly observed through changes in the UV–Vis absorption spectrum. When PIDF-BT was exposed to molecular dopants such as F4TCNQ (Appendix A) and F6TCNNQ (Appendix A), the original absorption of PIDF-BT almost disappeared, and polaron absorption formed by doping became prominent around 700 nm within 10 s of doping time. However, for Lewis acid dopants, although the changes in the absorption spectrum due to doping were similar to those of molecular dopants, the required doping time was more than 10 times longer to develop an absorption spectrum similar to that of PIDF-BT doped with molecular dopants. This result further implies that molecular dopants are more favorable for doping kinetics than Lewis acid dopants. In addition, the electrical conductivity of doped PIDF-BT varied significantly depending on the type of dopant applied (Figure 1c). Notably, high electrical conductivities exceeding 260 S/cm were achieved through doping with molecular dopants (F4TCNQ and F6TCNNQ), while AuCl_3_ and FeCl_3_ were the only effective dopants among the Lewis acids, resulting in similarly high electrical conductivities. Particularly, PIDF-BT doped with AuCl_3_ exhibited the highest electrical conductivity, exceeding 320 S/cm. Theoretical considerations suggest that the doping tendency is influenced by the energy offset between the highest occupied molecular orbital (HOMO) of the CP and the lowest unoccupied molecular orbital (LUMO) of the dopant [35,36]. To investigate this, the energy levels of each dopant were summarized [26,27,28,29,30,31], and the energy offset with PIDF-BT for each dopant was calculated by subtracting the LUMO of each dopant from the HOMO of PIDF-BT. However, no significant correlation was found between the energy offset and the electrical conductivity of doped PIDF-BT (Figure 1c and Appendix A), suggesting that while energy offset is a necessary factor for doping, it is not the sole determinant of doping efficiency [32].

The doping efficiency of the applied dopants and the resulting difference in electrical conductivity can be estimated from the UV–Vis absorption spectrum of doped PIDF-BT. Upon doping, the original absorption of PIDF-BT around 550 nm diminished, and a new absorption band emerged around 700 nm for all dopants (Appendix A), attributed to the absorption of polarons formed by doping [13]. Comparing the absorption intensity at 700 nm to that at 550 nm, AuCl_3_-doped PIDF-BT exhibited relatively high polaron absorption, correlating with its highest electrical conductivity. The trend in polaron intensity tended to decrease in the order of AuCl_3_, F4TCNQ (or F6TCNNQ), FeCl_3_, and NOBF_4_. The trend in relative intensity of polaron absorption mirrored the electrical conductivity of doped PIDF-BT, except for Magic Blue. Although Magic Blue showed absorption at approximately 700 nm comparable to AuCl_3_-doped PIDF-BT, its electrical conductivity was notably lower. This discrepancy in polaron absorption of Magic Blue could be attributed to its intrinsic absorption characteristics around 700 nm [37].

To enhance electrical conductivity further, PIDF-BT and CNT were dispersed in *o*-dichlorobenzene (DCB). Given that the electrical properties of CNTs significantly depend on their structural heterogeneity, we utilized a commercially available single-walled CNT (SWCNT, “TUBALL” with a diameter of ≤2 nm) to ensure the reproducibility of preparing PIDF-BT-wrapped SWCNT (PIDF-BT@SWCNT). As depicted in Figure 2a, when PIDF-BT and SWCNT are dispersed in DCB via tip sonication at 25 °C, PIDF-BT@SWCNT spontaneously forms in the DCB solution. Subsequently, massive SWCNT aggregates were removed through centrifugation, and the remaining DCB solution underwent filtration using a nylon membrane to eliminate excess PIDF-BT. This two-step purification process ensures the acquisition of PIDF-BT@SWCNT with excellent dispersion characteristics in DCB. Indeed, when a PIDF-BT@SWCNT buckypaper was redispersed in DCB after mechanically separating the nylon membrane filter, it demonstrated good dispersibility in DCB up to a concentration of 20 mg/mL, facilitating film formation through spin coating. Scanning electron microscopy (SEM) analysis of the film morphology of PIDF-BT@SWCNT revealed uniformly networked nanofibers, indicating successful wrapping of PIDF-BT around the surface of the SWCNT (Figure 2b). The wrapping of PIDF-BT on the SWCNT surface is evident from transmission electron microscopy (TEM) images. As shown in Figure 2c, individual SWCNT strands appear to have a diameter of approximately 2.0 nm, with PIDF-BT surrounding the outside of SWCNT, resulting in an overall diameter of the PIDF-BT@SWCNT strand in the range of 4.0 to 5.0 nm. The diameter of PIDF-BT@SWCNT observed in TEM is notably smaller than that of the nanofibers observed in SEM, likely due to the aggregation of PIDF-BT@SWCNT single strands during film formation, forming PIDF-BT@SWCNT bundles. Additionally, elemental analysis of the TEM image indicates uniform distribution of S, F, and N atoms, which are present only in PIDF-BT, along the SWCNT (Figure 2d,f). This result indirectly supports that PIDF-BT uniformly surrounds the SWCNT surface.

Based on the insights gained from the doping behavior of PIDF-BT, we evaluated the electrical conductivity of the prepared PIDF-BT@SWCNT by doping with molecular dopants (F4TCNQ and F6TCNNQ) for 10 s and Lewis acid dopants for 120 s, respectively. As depicted in Figure 3a, the initial electrical conductivity of pristine PIDF-BT@SWCNT was approximately 550 S/cm, but it dramatically increased upon doping, exceeding 2000 S/cm for all dopants. For example, PIDF-BT@SWCNT doped with AuCl_3_ achieved an electrical conductivity exceeding 4680 S/cm, with the electrical conductivities decreasing in the order of F4TCNQ (3230 S/cm), NOBF_4_ (2550 S/cm), FeCl_3_ (2460 S/cm), F6TCNNQ (2270 S/cm), and Magic Blue (2230 S/cm) dopants. Although AuCl_3_ doping resulted in the highest conductivity among the applied dopants, the electrical conductivity trend of doped PIDF-BT@SWCNT differed slightly from that of PIDF-BT doping, with the difference being particularly notable for Lewis dopants. This divergence in doping tendency between PIDF-BT and PIDF-BT@SWCNT with Lewis acid dopants is likely due to the potential doping ability of SWCNT with Lewis acid dopants [38,39,40], which can generate additional charge carriers in PIDF-BT@SWCNT in addition to those formed by doping of PIDF-BT surrounding SWCNT.

To further enhance the electrical conductivity of PIDF-BT@SWCNT, we conducted sequential cascade doping by combining different dopants. For cascade doping to be effective, the first dopant should have a shallower LUMO level than the second dopant [26]. Therefore, F4TCNQ, F6TCNNQ, and FeCl_3_ were chosen as the first dopants, while AuCl_3_, NOBF_4_, and Magic Blue were selected as the second dopants (Appendix A). In the dopant combinations of F4TCNQ/AuCl_3_ and F6TCNNQ/AuCl_3_, we achieved high electrical conductivity exceeding 6000 S/cm through cascade doping. The electrical conductivity of PIDF-BT@SWCNT doped using the F4TCNQ/AuCl_3_ combination was much higher than that of organic-based conductors, including CP, and was comparable to that of highly conductive ceramics such as MXene [41,42].

However, the increase in conductivity was negligible in the combination of other dopants compared to that of PIDF-BT@SWCNT doped with each single dopant (Figure 3b). To ensure high doping efficiency in cascade doping, it is crucial that the first dopant replaced by the second dopant should additionally dope peripheral CP chains and is swiftly replaced by the second dopant repeatedly [25]. From the doping kinetics of PIDF-BT, molecular dopants (F4TCNQ and F6TCNNQ) showed significantly faster doping kinetics than Lewis acid dopants, indicating that molecular dopants have an advantage as the first dopant in cascade doping. Hence, when a molecular dopant is used as the first dopant, higher doping efficiency can be expected than when FeCl_3_, a Lewis acid dopant, is used for cascade doping. Indeed, the FeCl_3_/AuCl_3_ dopant combination resulted in lower electrical conductivity than AuCl_3_ doping alone (Figure 3a). This can be attributed to the slow doping kinetics of FeCl_3_, which reduces the efficiency of dopant exchange and reintercalation into CP chains during cascade doping with the second dopant (AuCl_3_) [25]. Moreover, among the second dopants used with molecular dopants, only AuCl_3_ exhibited characteristically high electrical conductivity, likely due to its highest doping efficiency as a single dopant for PIDF-BT@SWCNT.

To understand the additional increase in electrical conductivity of PIDF-BT@SWCNT through sequential cascade doping, we compared the doping characteristics of F4TCNQ, AuCl_3_, and F4TCNQ/AuCl_3_ doping using UV–Vis absorption and ESR spectroscopy. As shown in Figure 4a, pristine PIDF-BT@SWCNT exhibited clear absorption between 400 and 600 nm, corresponding to the absorption of PIDF-BT, supporting successful SWCNT encapsulation. The absorption peak corresponding to PIDF-BT decreased with F4TCNQ doping and completely disappeared after AuCl_3_ doping, indicating the superior effectiveness of AuCl_3_. Upon F4TCNQ/AuCl_3_ cascade doping, the PIDF-BT-related absorption further diminished, and the polaron absorption at 700 nm slightly intensified compared to AuCl_3_ doping. The ESR spectrum also reflected these changes (Figure 4b). Pristine PIDF-BT@SWCNT showed a distinct ESR signal due to generated radicals during PIDF-BT wrapping of SWCNT [23]. F4TCNQ doping resulted in a characteristic ESR signal at a 2.003 g-factor, attributed to the F4TCNQ anion formed after PIDF-BT doping. However, this signal vanished in F4TCNQ/AuCl_3_ cascade doping, indicating efficient replacement of F4TCNQ by AuCl_3_. Moreover, the ESR intensity of F4TCNQ/AuCl_3_ cascade doping was slightly higher than that of AuCl_3_ alone, suggesting the generation of additional charge carriers and improved electrical conductivity. In summary, the results from UV–Vis absorption and ESR spectroscopy support the superior effectiveness of F4TCNQ/AuCl_3_ cascade doping compared to individual doping processes with F4TNCQ or AuCl_3_.

To elucidate the doping mechanism of PIDF-BT@SWCNT, we characterized the change in binding energies of the C, N, and S atoms involved in doping using XPS (Figure 5a). Compared to pristine PIDF-BT@SWCNT, the binding energies of C atoms remained unchanged, while the peaks of S and N atoms shifted to higher binding energies, indicating their electron-donating behavior upon doping. During F4TCNQ doping, we observed a shift in the S signals to higher binding energies, indicating their intercalation with the dopant. The N atom signal shifted both to higher and lower binding energies due to electron donation to F4TCNQ, simultaneously. In contrast, AuCl_3_ doping exhibited a more pronounced shift in the binding energies of N and S atoms towards higher energy regions compared to F4TCNQ doping. This trend was further enhanced in F4TCNQ/AuCl_3_ cascade doping, suggesting increased participation of N and S atoms in doping and charge carrier generation in the order of F4TCNQ, AuCl_3_, and F4TCNQ/AuCl_3_ doping. To quantitatively assess the amount of N and S atoms involved in doping at each step, we integrated the XPS signal to extract the relative proportions of N and S atoms interacting with the dopants. As shown in Figure 5b, initially, 18% of N and 11% of S atoms participated in F4TCNQ doping. This proportion significantly increased to 61% (N) and 18% (S) in AuCl_3_ doping, and further to 73% (N) and 30% (S) in F4TCNQ/AuCl_3_ doping. These results not only support the effectiveness of F4TCNQ/AuCl_3_ cascade doping but also suggest that N atoms are more advantageous for electron-donating type doping compared to S atoms.

To delve deeper into the doping characteristics of PIDF-BT@SWCNT, we compared the changes in Raman spectra of PIDF-BT@SWCNT with those of PIDF-BT at various doping stages (Figure 6a–c). Pristine PIDF-BT exhibited notable intensities at 1442 cm^−1^ and 1471 cm^−1^, corresponding to the C=C stretching from the thiophene and indoloindole units [43,44]. These characteristic Raman scatterings were similarly observed in pristine PIDF-BT@SWCNT, indicating successful encapsulation of SWCNT by PIDF-BT. Upon doping, the C=C stretching became broadened with a small redshift in both PIDF-BT and PIDF-BT@SWCNT due to the formation of quinoid structures from the benzenoid conjugated structures of PIDF-BT upon doping [38,45]. Quantification of the quinoid/benzenoid ratio through Gaussian fitting of the Raman scattering spectra (Appendix A) revealed an increase in the ratio in the order of F4TCNT, AuCl_3_, and F4TCNT/AuCl_3_ doping in both PIDF-BT and PIDF-BT@SWCNT (Figure 6d). As the quinoid structure in CP is formed by *p*-type doping [46,47], the increase in the quinoid structure ratio suggests that F4TCNT/AuCl_3_ cascade doping is more effective than using each dopant individually, leading to higher doping efficiency and, consequently, higher electrical conductivity. Additionally, Raman scattering bands not observed in PIDF-BT were detected at 1592 cm^−1^ (Figure 6b) and 2696 cm^−1^ (Figure 6c) in pristine PIDF-BT@SWCNT, corresponding to the G and G’ bands of SWCNT [38,39]. These characteristic G and G’ bands of SWCNTs also exhibited shifts upon doping. However, the G band shifted to higher wavenumbers (blueshift), contrary to the G’ band showing a redshift with doping. The blueshift of the G band could be attributed to increased vibration energy due to electron shielding effects from *p*-type doping [38,39]. Contrastingly, the redshifting tendency of the G’ band may result from increased shear compression on the core SWCNT by PIDF-BT, which expands through dopant intercalation after doping [23,48].

## 3. Materials and Methods

### 3.1. Materials

All chemicals used in this study were purchased from commercial suppliers (Sigma-Aldrich Korea and Tokyo Chemical Industry (SEJIN CI, Gangnam-Gu, Republic of Korea)) and used without further purification. SWCNTs (“TUBALL”, diameter ≤ 2 nm) and F4TCNQ were obtained from OCSiAL and OSSILA, respectively, and used as received. PIDF-BT was synthesized as previously reported [49,50].

### 3.2. PIDF-BT@SWCNT Preparation

PIDF-BT and SWCNT were suspended in *o*-dichlorobenzene (DCB, 1.5 mg/mL) at a 1:2 weight ratio. The solution was then tip-sonicated at 130 W for 30 min and subsequently stirred for 12 h at 60 °C. Massive SWCNT aggregates were removed through centrifugation (Hitachi (Tokyo, Japan), CR22GIII) at 18,000 rpm for 30 min. The resulting supernatant was filtered through a nylon membrane filter (Whatman (Shrewsbury, MA, USA), pore size 0.2 μm) to remove excess PIDF-BT. To eliminate unwrapped PIDF-BT, the membrane filter was rinsed with warm DCB until the filtrate became transparent. The membrane was then dried in a vacuum oven for 12 h, and a PIDF-BT@SWCNT buckypaper was obtained by mechanically separating the membrane.

### 3.3. Sequential Doping and Conductivity Measurement

PIDF-BT was dissolved in DCB (5 mg/mL), and thin films (approximately 100 μm) were prepared on glass substrates through spin coating at 1000 rpm. The films were immersed in a doping solution (0.5 wt%) at 60 °C, and the immersion time was adjusted to estimate the doping kinetics of each dopant. For F4TCNQ and F6TCNNQ, the dopants were dissolved in acetone, while other dopants were dissolved in acetonitrile. The doped PIDF-BT films were gently dried under nitrogen, and electrical conductivity was calculated from the sheet resistance measured using a 4-point probe resist meter (Nittoseiko Analytech (Kanagawa, Japan), Loresta-GX MCP-T700). The electrical conductivity of the doped PIDF-BT@SWCNT was measured following the same procedure as for PIDF-BT after preparing the PIDF-BT@SWCNT film. The PIDF-BT@SWCNT buckypaper was dispersed at 25 °C for 20 min in a DCB: chloroform mixture (1:2, 3.0 mg/mL) using bath sonication (HWASHIN Tech (Daegu, Republic of Korea), Power Sonic 410) and subsequently stirred at 60 °C for 1 h. A PIDF-BT@SWCNT thin film (approximately 100 μm) was prepared through spin coating of the dispersion at 1000 rpm.

### 3.4. Characterization

The absorption spectra of PIDF-BT and PIDF-BT@SWCNT were analyzed before and after doping using a UV–Vis spectrometer (JASCO (Tokyo, Japan), V-670). The ESR spectra were acquired using an EMXplus-9.5/12/P/L system (Bruker, Pangyo-ro, Republic of Korea). Alterations in atomic binding energy for representative atoms (C, S, and N) at each doping step were investigated via XPS using a Thermo Fisher Scientific K-Alpha instrument. Shifts in Raman spectra upon doping were assessed using a spectrophotometer (inVia Reflex, Renishaw, Gyeonggi-do, Republic of Korea) with a 514 nm laser source.

## 4. Conclusions

To create a highly conductive and flexible platform, we prepared PIDF-BT@SWCNT by encapsulating SWCNT with PIDF-BT, known for its effective doping properties. Through a systematic evaluation of the doping behavior of PIDF-BT with various dopants, we identified suitable dopant combinations for cascade doping to enhance the doping efficiency of PIDF-BT@SWCNT. Utilizing F4TCNQ and F6TCNNQ, known for their rapid doping kinetics, as the first dopants, and AuCl_3_, recognized for its high doping efficiency as a single dopant, as the second dopant in cascade doping significantly improved the doping efficiency of PIDF-BT@SWCNT. Therefore, the cascade doping with F4TCNQ followed by AuCl_3_ resulted in significantly higher electrical conductivity, exceeding 6000 S/cm, compared to conductivity levels achieved with F4TCNQ (3200 S/cm) or AuCl_3_ (4600 S/cm) alone. The doping mechanism of PIDF-BT@SWCNT in the F4TCNQ/AuCl_3_ cascade doping configuration was comprehensively characterized using XPS and Raman spectroscopy. Our findings revealed that the proportion of electron-donating atoms involved in doping and the ratio of quinoid structures generated by doping increased sequentially with F4TCNQ, AuCl_3_, and F4TCNQ/AuCl_3_ doping. These results underscore the efficacy of cascade doping and provide valuable insights into optimizing dopant combinations to achieve high doping efficiency. Furthermore, our study highlights the advantages of cascade doping in terms of doping efficiency and resulting electrical conductivity, compared to single-dopant processes, when dopant combinations are carefully controlled. These findings offer valuable strategies for enhancing the performance of flexible and highly conductive platforms like PIDF-BT@SWCNT through cascade doping.

## Figures and Tables

**Figure 1 polymers-16-01884-f001:**
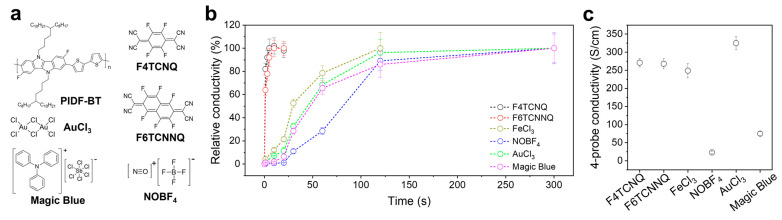
(**a**) Chemical structures of PIDF-BT and dopants, (**b**) changes in electrical conductivity according to doping time, and (**c**) 4−probe electrical conductivity of doped PIDF-BT with each dopant.

**Figure 2 polymers-16-01884-f002:**
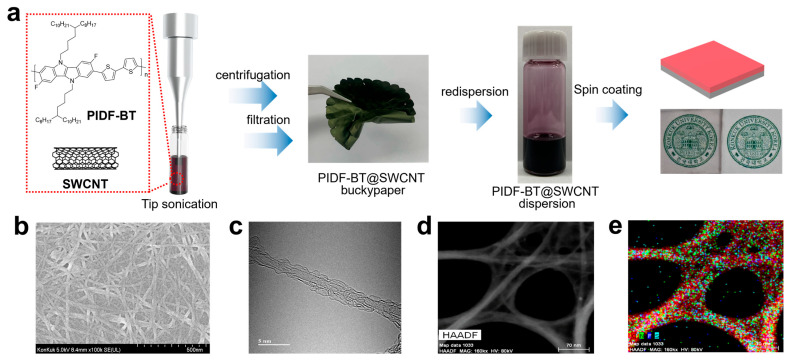
(**a**) Schematic of PIDF-BT@SWCNT preparation, including optical transparency, (**b**) SEM image of the PIDF-BT@SWCNT film, TEM image of (**c**) a single strand and (**d**) bundled PIDF-BT@SWCNT, and (**e**) atomic distribution on the bundled PIDF-BT@SWCNT.

**Figure 3 polymers-16-01884-f003:**
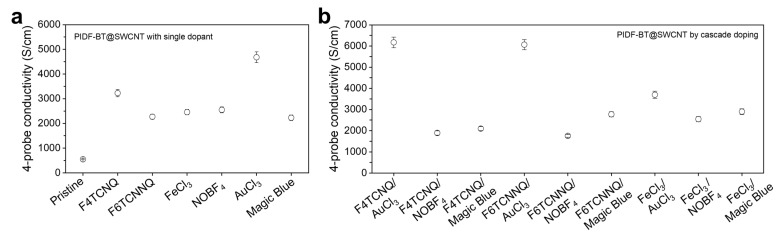
4-probe electrical conductivity of doped PIDF-BT@SWCNT through (**a**) sequential doping with single dopants and (**b**) sequential cascade doping with combined dopants.

**Figure 4 polymers-16-01884-f004:**
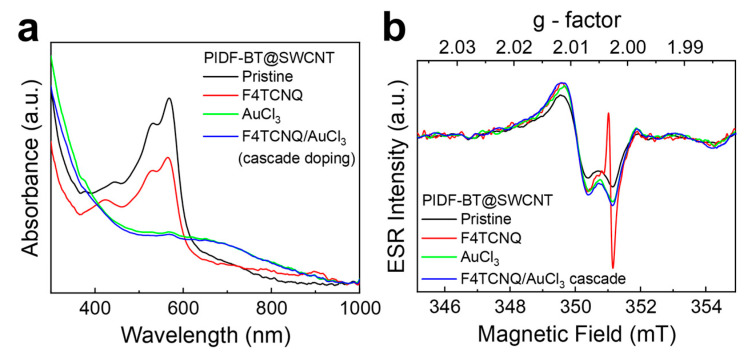
(**a**) Change in UV–Vis spectra of PIDF-BT@SWCNT before and after doping, and (**b**) ESR spectra of doped PIDF-BT@SWCNT.

**Figure 5 polymers-16-01884-f005:**
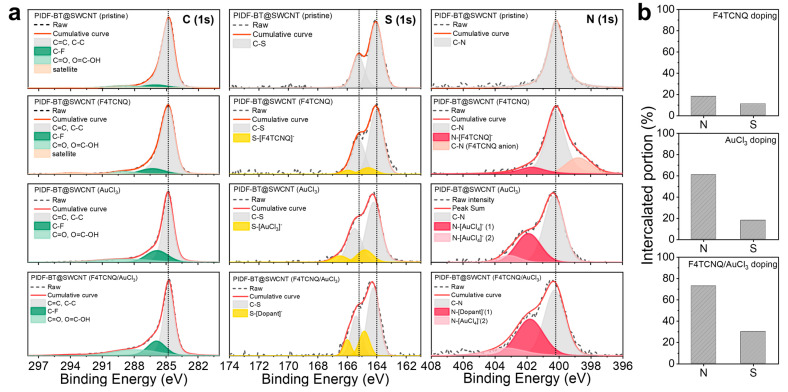
(**a**) XPS atomic binding energies of C, S, and N atoms in PIDF-BT@SWCNT, and (**b**) their relative intercalated portions in each doping stage during F4TCNQ/AuCl_3_ cascade doping.

**Figure 6 polymers-16-01884-f006:**
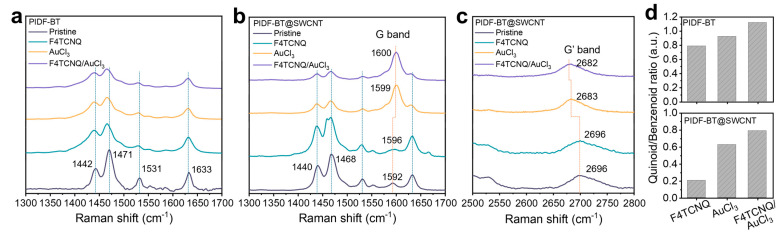
(**a**) Raman spectra of PIDF-BT and (**b**,**c**) PIDF-BT@SWCNT before and after doping, and (**d**) quinoid/benzenoid relative ratio of PIDF-BT and PIDF-BT@SWCNT at each doping stage.

## Data Availability

Data are contained within the article and Appendix A.

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
