# Peer review of "Sequential Cascade Doping of Conjugated-Polymer-Wrapped Carbon Nanotubes for Highly Electrically Conductive Platforms"

_polymers, 2024, doi:10.3390/polym16131884_

Round 1

Reviewer 1 Report

Comments and Suggestions for Authors

The authors developed a work involving sequential doping of conjugated polymer-wrapped carbon nanotubes in order to increase the electrical conductivity of this material for possible applications in flexible conductive platforms. The text was fluid and well written, the research topic appears to be very current, interesting, and promising, and the results were well discussed. However, there are some problems that must be resolved until the publication of the paper:

- The keyword ‘doping mechanism’ appears twice;

- The paragraph “It was observed … alone” (lines 81-84) should not appear in the introduction but in the abstract or results and discussion;

- The compound NOBF4 appears several times without the 4 subscripts (examples: line 97, fig 1a and 3, lines 145, 196, 209, etc), as well as the compound AuCl3 at the conclusions (lines 352, 356, 359);

- Demonstrate that the summarized data is contained in Fig. S2a when cited in line 133;

- It is mL for milliliters, not ml, as appears in line 170, 314, 324, 334;

- As mentioned in lines 206 and 207, would it be possible to prepare other combination systems like FeCl3/F4TCNQ, FeCl3/F6TCNNQ, or F4TCNQ/F6TCNNQ? Likewise, a more complex system F4TCNQ/F6TCNNQ/AuCl3 be possible? This explanation could improve the discussion of the results;

- I suggest adding references to the end of the paragraph on lines 222-224 to support the statement;

- One reference is missing in line 299;

- Italic in p-type, line 298;

- There is no acknowledgment

Comments on the Quality of English Language

No additional comments.

Author Response

Reviewer 1

The authors developed a work involving sequential doping of conjugated polymer-wrapped carbon nanotubes in order to increase the electrical conductivity of this material for possible applications in flexible conductive platforms. The text was fluid and well written, the research topic appears to be very current, interesting, and promising, and the results were well discussed. However, there are some problems that must be resolved until the publication of the paper:

  1. The keyword ‘doping mechanism’ appears twice

Answer: Thanks to the reviewer for their kindness. One ‘doping mechanism’ keyword was replaced with ‘flexible conductor’ in the keyword section.

  1. The paragraph “It was observed … alone” (lines 81-84) should not appear in the introduction but in the abstract or results and discussion;

Answer: As suggested by the reviewer, the paragraph was removed from the introduction and rearranged in the conclusion part.

  1. The compound NOBF4 appears several times without the 4 subscripts (examples: line 97, fig 1a and 3, lines 145, 196, 209, etc), as well as the compound AuCl3 at the conclusions (lines 352, 356, 359);

Answer: Thanks to the reviewer for their kindness. All typos have been fixed.

  1. Demonstrate that the summarized data is contained in Fig. S2a when cited in line 133;

Answer: As suggested by the reviewer, references providing information in Figure S2a have been indicated.

  1. It is mL for milliliters, not ml, as appears in line 170, 314, 324, 334;

Answer: Thanks to the reviewer for their kindness. All typos have been fixed.

  1. As mentioned in lines 206 and 207, would it be possible to prepare other combination systems like FeCl3/F4TCNQ, FeCl3/F6TCNNQ, or F4TCNQ/F6TCNNQ? Likewise, a more complex system F4TCNQ/F6TCNNQ/AuCl3 be possible? This explanation could improve the discussion of the results;

Answer: As described in lines 204-214, for effective cascade doping, the first dopant should have a shallower LUMO level than the second dopant to gain an energetic driving forces. Therefore, F4TCNQ, F6TCNNQ and FeCl3, which have relatively shallow LUMO levels, were selected as the first dopant for the cascade doping. In addition, from Figure 3b, the importance of the doping kinetics of the first dopant can be confirmed. Conclusively, this manuscript not only provides an excellent electrically conductive platform but also offers a dopant selection strategy for effective cascade doping. Indeed, it is difficult to expect an energetic gain from the FeCl3/F4TCNQ combination, and FeCl3/F6TCNNQ is not beneficial from a kinetics perspective. In the case of the F4TCNQ/F6TCNNQ combination, the electrical conductivity was about 3000 S/cm, which did not reach that of F4TCNQ single doping. The poor electrical conductivity of F4TCNQ/F6TCNNQ could be ascribed to the competitive doping kinetics between F4TCNQ and F6TCNNQ. Although the F4TCNQ/F6TCNNQ/AuCl3 combination is interesting, drawing clear conclusions may prove challenging.

  1. I suggest adding references to the end of the paragraph on lines 222-224 to support the statement;

Answer: As suggested by the reviewer, a reference (Adv. Mater. 2020, 32, 200512) has been added to the paragraph to support the contribution of doping kinetics during cascade doping.

  1. One reference is missing in line 299;

Answer: Thanks to the reviewer for their kindness. The reference was added.

  1. Italic in p-type, line 298;

Answer: Thanks to the reviewer for their kindness. The word was edited to italics.

  1. There is no acknowledgment

Answer: There is one acknowledgment in the original manuscript: This paper was written as part of Konkuk University's research support program for its faculty on sabbatical leave in 2023.

Reviewer 2 Report

Comments and Suggestions for Authors

In this work, the authors develops PIDF-BT@SWCNT by wrapping single-walled carbon nanotubes (SWCNTs) with a conjugated polymer, which demonstrates remarkable doping efficiency, surpassing that of the individual dopants and yielding an exceptional electrical conductivity exceeding 6,000 18 S/cm. Based on this result, I suggest accept it after some revision:

(1) FTIR data should be added.

(2) Performance comparison with other works should be added.

(3) Preparation process of SWCNT should be added.

(4) Doping mechanism should be provided.

(5) Some CNT-related papers such as 10.1007/s42765-023-00312-5, 10.1016/j.jcis.2023.04.076, 10.1039/D0TA04943C, etc should be discussed in Introduction part.

Author Response

Reviewer 2

In this work, the authors develops PIDF-BT@SWCNT by wrapping single-walled carbon nanotubes (SWCNTs) with a conjugated polymer, which demonstrates remarkable doping efficiency, surpassing that of the individual dopants and yielding an exceptional electrical conductivity exceeding 6,000 18 S/cm. Based on this result, I suggest accept it after some revision:

(1) FTIR data should be added.

Answer: We appreciate this meaningful suggestion. However, we believe that the doping behavior and mechanism of PIDF-BT@SWCNT can be adequately elucidated using the XPS and Raman data enclosed in the original manuscript.

(2) Performance comparison with other works should be added.

Answer: As suggested by the reviewer, the superior electrical conductivity of doped PIDF-BT@SWCNTs was briefly compared with that of other solution-processable flexible conductors (page 5) as follows:

“The electrical conductivity of PIDF-BT@SWCNT doped using the F4TCNQ/AuCl3 combination was much higher than that of organic-based conductors, including CP, and was comparable to that of highly conductive ceramics such as MXene”

(3) Preparation process of SWCNT should be added.

Answer: As described in lines 157-160, we utilized a commercially available single-walled CNT (SWCNT, TUBALL with a diameter of ≤2 nm) to ensure the reproducibility of preparing PIDF-BT-wrapped SWCNT (PIDF-BT@SWCNT).

(4) Doping mechanism should be provided.

Answer: The general doping mechanism of PIDF-BT@SWCNT for cascade doping is described from the bottom of page 5 to the top of page 6 in the revised manuscript. Additionally, the detailed doping mechanism of PIDF-BT@SWCNT for F4TCNQ/AuCl3 cascade doping is elucidated using XPS, Raman, and ESR analysis.

(5) Some CNT-related papers such as 10.1007/s42765-023-00312-5, 10.1016/j.jcis.2023.04.076, 10.1039/D0TA04943C, etc should be discussed in Introduction part.

Answer: The suggested papers were included in the introduction as references 20-22.

Reviewer 3 Report

Comments and Suggestions for Authors

In the manuscript, the authors have explored the feasibility of processes aimed at improving the electrical doping of polymer-wrapped single-walled carbon nanotubes to obtain high electrical conductivity. Several dopant substances have been considered and compared by testing the electrical properties of samples. The study was supported by UV-vis spectroscopy, Raman spectroscopy, XRS and ESR analysis.

The experiments gave an interesting insight into the doping processes of polymers, adding information of large interest on the fabrication of transparent and high-conductive electronic materials.

The processes and characterization were clearly described and the references are adequate. For these reasons, I recommend the publication of this manuscript with some minor adjustments here listed :

at line 161: indicate that the term “TUBALL” is a Trademark.

at line 281: Please, define “IDID”

at line 282 and line 293: “absorption” is not a correct term when referring to Raman spectroscopy, which is based on a scattering process of light.

at line 299: the reference is missing.

At lines 296-301: The shift of mode energies in the Raman spectrum of the graphene-based materials can also be related to doping changes of the graphene itself (see, for instance, A. Das et al, Nature nanotechnology 3 (2008) 210-215). The reported spectrum changes are compatible with an increased electron concentration of doped graphene. Could the doping process affect also the carbon nanotubes properties? A comment on this aspect is required.

At lines 352, 359: correct subscripts.

Author Response

Reviewer 3

In the manuscript, the authors have explored the feasibility of processes aimed at improving the electrical doping of polymer-wrapped single-walled carbon nanotubes to obtain high electrical conductivity. Several dopant substances have been considered and compared by testing the electrical properties of samples. The study was supported by UV-vis spectroscopy, Raman spectroscopy, XRS and ESR analysis.

The experiments gave an interesting insight into the doping processes of polymers, adding information of large interest on the fabrication of transparent and high-conductive electronic materials.

The processes and characterization were clearly described and the references are adequate. For these reasons, I recommend the publication of this manuscript with some minor adjustments here listed :

  1. at line 161: indicate that the term “TUBALL” is a Trademark.

Answer: Thank to the reviewer for their kindness. “TUBALL” was marked as suggested.

  1. at line 281: Please, define “IDID”

Answer: Thank to the reviewer for their kindness. The “IDID” was replaced with “indoloindole”.

  1. at line 282 and line 293: “absorption” is not a correct term when referring to Raman spectroscopy, which is based on a scattering process of light.

Answer: Thank to the reviewer for their kindness. “absorption” was modified to “scattering”.

  1. at line 299: the reference is missing.

Answer: Thanks to the reviewer for their kindness. The reference was added.

  1. At lines 296-301: The shift of mode energies in the Raman spectrum of the graphene-based materials can also be related to doping changes of the graphene itself (see, for instance, A. Das et al, Nature nanotechnology 3 (2008) 210-215). The reported spectrum changes are compatible with an increased electron concentration of doped graphene. Could the doping process affect also the carbon nanotubes properties? A comment on this aspect is required.

Answer: It is well-known that when SWCNTs are doped with p-type dopants, atomic vibrations are screened by electrons, resulting in an increase in vibration energy and a blueshift of the G and G' bands (J. Am. Chem. Soc. 2008, 130, 12757; ACS Nano 2011, 5, 1236). Specifically, the G and G' bands tend to shift in the same direction as changes in electron density in SWCNTs.

Therefore, unlike graphene, the opposite shift trend of the G and G' bands upon p-type doping observed in this study is not solely determined by changes in electron density. It is also well-known that when SWCNTs are subjected to shear strain, the G' band redshifts (Compos. Part A: Appl. 2001, 32, 401). Since conjugated polymer chains wrapping SWCNTs induce shear strain through dopant intercalation within polymer chains during doping, it is plausible that the opposite shift trend of the G and G' bands in this study could be attributed to the complex effects of changes in electron density and mechanical stress caused by doping

  1. At lines 352, 359: correct subscripts.

Answer: Thanks to the reviewer for their kindness. All typos have been fixed.

Reviewer 4 Report

Comments and Suggestions for Authors

In the manuscript entitled „Sequential Cascade Doping of Conjugated-Polymer-Wrapped Carbon Nanotubes for Highly Electrically Conductive Plat-3 forms” authors show results of cascade doping of the p-type conjugated polymer, poly(2-([2,20-bithiophen]-5-yl)-3,8-difluoro-5,10-bis(5-octylpentadecyl)-5,10-dihydroindolo [3,2-b] indole (PIDF-BT), wrapped on single-walled carbon nanotubes (SWCNTs). They have shown that doping with 2,3,5,6-tetrafluoro-7,7,8,8-tetracyanoquinodimethane (F4-TCNQ) or F6-TCNNQ followed with the second doping with AuCl3 results in electrical conductivity that exceeds 6000 S/cm. This value is only two orders of magnitude lower than silver and copper. Obtained results are supported by extensive characterization including conductivity measurements, UV-Vis absorption, electron microscopy with elemental analysis, Raman spectroscopy and ESR.

Apart from the fact of obtaining extraordinary electrical conductivity, it is important to emphasize the high competence of the team, which is expressed in the skillful selection of research methods that allowed a deeper understanding of the phenomena related to charge transfer in the studied system, which affected the conductivity.

The results obtained, combined with the methodology adopted, are sure to be of great interest to readers, as well as the article will have a great impact on further research on doping strategies of conjugated polymers. Therefore, I confidently recommend the manuscript for publication as it is.

Some minor comments on typos:

(1)  Figure 2a, line 170 and 325: I think it is more about ‘spin coating’ instead of ‘spin casting’.

(2)  Line 287: ‘Figure S3’ instead of ‘Figure S4’.

(3)  Line 311: ‘TUBALL’ instead of ‘TUBAL’ and ‘OCSiAl’ instead of ‘OSCiAL’.

Author Response

Reviewer 4

In the manuscript entitled „Sequential Cascade Doping of Conjugated-Polymer-Wrapped Carbon Nanotubes for Highly Electrically Conductive Plat-3 forms” authors show results of cascade doping of the p-type conjugated polymer, poly(2-([2,20-bithiophen]-5-yl)-3,8-difluoro-5,10-bis(5-octylpentadecyl)-5,10-dihydroindolo [3,2-b] indole (PIDF-BT), wrapped on single-walled carbon nanotubes (SWCNTs). They have shown that doping with 2,3,5,6-tetrafluoro-7,7,8,8-tetracyanoquinodimethane (F4-TCNQ) or F6-TCNNQ followed with the second doping with AuCl3 results in electrical conductivity that exceeds 6000 S/cm. This value is only two orders of magnitude lower than silver and copper. Obtained results are supported by extensive characterization including conductivity measurements, UV-Vis absorption, electron microscopy with elemental analysis, Raman spectroscopy and ESR.

Apart from the fact of obtaining extraordinary electrical conductivity, it is important to emphasize the high competence of the team, which is expressed in the skillful selection of research methods that allowed a deeper understanding of the phenomena related to charge transfer in the studied system, which affected the conductivity.

The results obtained, combined with the methodology adopted, are sure to be of great interest to readers, as well as the article will have a great impact on further research on doping strategies of conjugated polymers. Therefore, I confidently recommend the manuscript for publication as it is.

Some minor comments on typos:

(1)  Figure 2a, line 170 and 325: I think it is more about ‘spin coating’ instead of ‘spin casting’.

Answer: Thanks to the reviewer for their kindness. As suggested, ‘spin casting’ was replaced with ‘spin coating’.

(2)  Line 287: ‘Figure S3’ instead of ‘Figure S4’.

Answer: Thanks to the reviewer for their kindness. The mistake was corrected.

(3)  Line 311: ‘TUBALL’ instead of ‘TUBAL’ and ‘OCSiAl’ instead of ‘OSCiAL’.

Answer: Thanks to the reviewer for their kindness. All typos have been fixed.

Round 2

Reviewer 2 Report

Comments and Suggestions for Authors

This version is nice, please accept it.